



# Global impact of nitrate photolysis in sea-salt aerosol on NOx, OH, and O3 in the marine boundary layer

Prasad Kasibhatla[1], Tomás Sherwen[2,3], Mathew J. Evans[2,3], Lucy J. Carpenter[2], Chris Reed[2,4], Becky Alexander[5], Qianjie Chen[5], Melissa Sulprizio[6], James D. Lee[2], Katie A. Read[2], William Bloss[7], Leigh R. Crilley[7], William C. Keene[8], Alex A. P. Pszenny[9], Alma Hodzic[10,11]

[1]Nicholas School of the Environment, Duke University, Durham, NC 27708, USA
[2]Wolfson Atmospheric Chemistry Laboratories, Department of Chemistry, University of York, York YO10 5DD, UK
[3]National Centre for Atmospheric Science, University of York, York YO10 5DD, UK
[4]Facility for Airborne Atmospheric Measurements (FAAM), Cranfield University, Cranfield, MK43 0AL, UK
[5]Department of Atmospheric Sciences, University of Washington, Seattle, WA 98195, USA
[6]School of Engineering and Applied Sciences, Harvard University, Cambridge, MA 02138, USA
[7]School of Geography, Earth and Environmental Sciences, University of Birmingham, Edgbaston, Birmingham, B15 2TT, UK
[8]Department of Environmental Sciences, University of Virginia, Charlottesville, VA, 22904, USA
[9]Institute for the Study of Earth, Oceans, and Space, University of New Hampshire, Durham, NH 03824, USA
[10]National Center for Atmospheric Research, Boulder, CO 80305, USA
[11]Laboratoire d'Aérologie, Université de Toulouse, CNRS, University Paul Sabatier, France

*Correspondence to*: Prasad Kasibhatla (psk9@duke.edu)

**Abstract.** Recent field studies have suggested that sea-salt particulate nitrate (NITs) photolysis may act as a significant local source of nitrogen oxides (NOx) over oceans. We present a study of the global impact of this process on oxidant concentrations in the marine boundary layer using the GEOS-Chem model, after first updating the model to better simulate observed gas/particle phase partitioning of nitrate in the marine boundary layer. Model comparisons with long-term measurements of NOx from the Cape Verde Atmospheric Observatory (CVAO) in the eastern tropical North Atlantic provide support for an *in situ* source of NOx from NITs photolysis, with NITs photolysis coefficients about 25-50 times larger than corresponding HNO3 photolysis coefficients. Short-term measurement of nitrous acid (HONO) at this location show a clear daytime peak, with average peak mixing ratios ranging from 3 to 6 pptv. The model reproduces the general shape of the diurnal HONO profile only when NITs photolysis is included, but the magnitude of the daytime peak mixing ratio is under-predicted. This under-prediction is somewhat reduced if HONO yields from NITs photolysis are assumed to be close to unity. The combined NOx and HONO analysis suggests that the upper limit of the ratio of NITs:HNO3 photolysis coefficients is about 100. The largest simulated relative impact of NITs photolysis is in the tropical and subtropical marine boundary layer, with peak local enhancements ranging from factors of 5-20 for NOx, 1.2-1.6 for OH, and 1.1-1.3 for ozone. Since the spatial extent of the sea-salt aerosol impact is limited, global impacts on NOx, ozone and OH mass burdens are small (~1-3 %). We also present preliminary analysis showing that particulate nitrate photolysis in accumulation-mode aerosols (predominantly over continental regions) could lead to ppbv-level increases in ozone in the continental boundary layer. Our results highlight the



need for more comprehensive long-term measurements of $NO_x$, and related species like HONO and sea-salt particulate nitrate, to better constrain the impact of particulate nitrate photolysis on marine boundary layer oxidant chemistry. Further field and laboratory studies on particulate nitrate photolysis in other aerosol types are also needed to better understand the impact of this process on continental boundary layer oxidant chemistry.

**1 Introduction**

Nitrogen oxides ($NO_x$) are a key component of tropospheric chemistry due to their impact on oxidant chemical cycles. This paper focuses on the $NO_x$ budget of the marine boundary layer (MBL). The traditional view has been that relatively low concentrations of $NO_x$ in the MBL are maintained by a balance between the transport of $NO_x$ to the MBL (either by direct transport of $NO_x$ from continental source regions or by transport and subsequent decomposition of $NO_x$ reservoirs such as

organic nitrates) and chemical destruction by reaction with hydroxyl radicals (OH) (e.g., Moxim et al., 1996). $NO_x$ emissions from ships (Lawrence and Crutzen, 1999; Kasibhatla et al., 2000; Beirle et al., 2004), downward transport of lightning-produced $NO_x$ from the free troposphere (Levy et al., 1996, 1999), and halogen-mediated chemistry (von Glasow et al., 2004; Keene et al., 2009; Long et al., 2014; Schmidt et al., 2016; Sherwen et al., 2016a, 2016b) also modulate $NO_x$ levels in the MBL.

Past laboratory studies have suggested that recycling of $NO_x$ by photochemical reactions of adsorbed nitrate on surfaces might also serve as an *in situ* $NO_x$ source (e.g. Zhou et al., 2003; Handley et al. 2007; Chen et al., 2011; Laufs and Kleffmann, 2016). More recently, field studies have suggested that photolysis of particulate nitrate (p-$NO_3$) associated with MBL sea-salt aerosols (SSA) can be a significant source of $NO_x$. In particular, Ye et al. (2016) analyzed trace gas measurements from two aircraft

flights over the western subtropical North Atlantic Ocean during the summer 2013 Nitrogen, Oxidants, Mercury and Aerosol Distributions, Sources and Sinks (NOMADSS) field campaign and hypothesized that measured daytime nitrous acid (HONO) concentrations in the MBL were indicative of a significant *in situ* source from p-$NO_3$ photolysis (since HONO photolyses rapidly to yield $NO_x$, p-$NO_3$ photolysis is effectively a $NO_x$ source). Their box model analysis suggested that the peak p-$NO_3$ photolysis rate coefficient was of the order of $2 \times 10^{-4}$ $s^{-1}$, about 300 times larger than the corresponding photolysis coefficient

for gas-phase nitric acid ($HNO_3$). Manipulation experiments involving irradiation of samples of ambient (Ye et al., 2016, 2017b) and synthetic marine aerosol (Zhou et al., 2008) provide additional support for the hypothesized importance of p-$NO_3$ photolysis in the MBL. It is also worth noting that photolysis of nitrate in seawater has been measured at naturally occurring concentrations, albeit at much lower rates with photolysis rate coefficients averaging $2 \times 10^{-8}$ $s^{-1}$ (Zafiriou and True, 1979).

Measurements of $NO_x$ and HONO at the Cape Verde Observatory in the tropical eastern Atlantic Ocean (CVAO, 16.864° N, 24.868° W) have provided additional evidence for *in situ* $NO_x$ production in the MBL (Reed et al., 2017). Long-term $NO_x$ measurements at CVAO show a consistent diurnal cycle, with peak daytime mixing ratios 10-15 pptv higher than night-time



levels in all seasons except in winter (Reed et al., 2017). Shorter term HONO measurements from CVAO also show a strong diurnal cycle, peaking at a few pptv around midday and dropping to near zero at night (Reed et al., 2017). The observed levels of HONO again indicate an *in situ* source, while the diurnal cycle suggests that this source is photolytic. Box model analysis of the CVAO measurements indeed indicate that rapid recycling of $NO_x$ via p-$NO_3$ photolysis is an important source of $NO_x$

in the tropical MBL, though estimated p-$NO_3$ photolysis rate coefficients were only 10-15 times larger than $HNO_3$ photolysis rate coefficients. (Reed et al., 2017; Ye et al., 2017). While some of the variability in estimated p-$NO_3$ photolysis rates between the NOMADSS and CVAO analyses may be due to differences in SSA composition at the two locations (Ye et al., 2017), the limited measurements preclude from definitive conclusions in this regard.

In this paper, we explore the global-scale significance of this newly-recognized p-$NO_3$ photolysis mechanism for tropospheric oxidant chemistry using the GEOS-Chem global chemical transport model. We perform a series of simulations to assess the sensitivity of $NO_x$ to the rate of p-$NO_3$ photolysis in SSA in the MBL, and attempt to constrain the magnitude of this *in situ* $NO_x$ source using $NO_x$ and HONO measurements from CVAO. We then estimate the impact of p-$NO_3$ photolysis on the large-scale distribution of $NO_x$, $O_3$ and OH in the MBL. Finally, we perform an initial analysis of the impact of photolysis of nitrate

on other aerosol types and consider its implications for continental boundary layer oxidant chemistry.

## 2 Models and measurements

### 2.1 GEOS-Chem model configuration

We use the GEOS-Chem chemical transport model (v10-01, Bey et al., 2001; www.geos-chem.org), as updated by Sherwen et al. (2017). The model includes a comprehensive tropospheric $NO_x$-VOC-$HO_x$-$O_x$ oxidant chemistry scheme (Mao et al.,

2013), coupled to an aerosol module that includes inorganic sulfate, ammonium, and nitrate aerosols (Park et al., 2004), as well as primary black carbon, organic carbon, dust, and sea-salt aerosols (Park et al, 2003; Fairlie et al., 2007; Alexander et al., 2005). The oxidant chemistry scheme also includes a detailed treatment of tropospheric halogen chemistry (Parrella et al., 2012; Schmidt et al., 2016; Sherwen et al. 2016a, 2016b, 2017). Trace gas and aerosol emissions are specified as described in Ford and Heald (2012).

The model transports two size classes of SSA, representing accumulation-mode (0.01-0.5 μm dry radius) and coarse-mode (0.5-8 μm dry radius) SSA. Emissions of SSA are calculated online in the model using the formulation of Gong et al. (2003), as described by Jaeglé et al. (2011). The model explicitly tracks particulate sulfate and nitrate associated with coarse-mode SSA (identified as "$SO_4s$"and "NITs", respectively, in the model), with $SO_4s$ production and gas-phase $HNO_3$ uptake (to form

NITs) calculated as described by Alexander et al. (2005). In this scheme, $SO_4s$ and NITs formation at each time step is determined by the mass flux of gas-phase sulfur dioxide and $HNO_3$ to coarse-mode SSA, but limited by the local alkalinity



emission flux in that time step. In Section 3.1, we show that this imposed alkalinity limitation unrealistically limits NITs

production in the MBL. We therefore update the model calculation of HNO₃ uptake on coarse-mode SSA as described below.

Laboratory measurements (e.g. Guimbaud et al., 2002) and field studies (e.g. Keene et al., 2009) indicate that freshly produced

SSA are rapidly acidified via incorporation of HNO₃ and other acids leading to the volatilization of hydrochloric acid in

accordance with expectations based on respective thermodynamic properties (Henry's Law and dissociation constants) and

associated meteorological conditions (relative humidity and temperature). We therefore updated the model to calculate HNO₃

uptake (and concomitant NITs production) by coarse-mode SSA using the mass transfer equation appropriate for the transition

regime as in Feng and Penner (2007):

$$-\frac{d}{dt}[HNO_3] = \frac{d}{dt}[NITs] = 4\,\pi\,D_g\,R_{cSSA}\,N_{cSSA}\,f(Kn)\,([HNO_3] - [HNO_3]^{eq})$$    (1)

where, $[.]$ represents mixing ratio, $D_g$ is the gas-phase diffusivity of HNO₃, $R_{cSSA}$ is the radius of coarse-mode SSA, $N_{cSSA}$ is

the number concentration of coarse-mode SSA, and Kn is the Knudsen number. $[HNO_3]^{eq}$ is the equilibrium concentration of

gas-phase HNO₃ associated with coarse-mode SSA, determined using ISORROPIA II (Fountoukis and Nenes, 2007). Since

the model does not include aerosol microphysics, $R_{cSSA}$ and $N_{cSSA}$ are estimated from effective dry radius and mass

concentration of coarse-mode SSA, assuming a dry SSA density of 2200 kg m⁻³ and using relative humidity dependent SSA

growth factors (Lewis and Schwartz, 2006). The function $f(Kn)$ is the ratio of gas-phase flux to the particle in the transition

region to that in the continuum regime, and is specified using the Fuchs and Sutugin (1971) formulation:

$$f(Kn) = \frac{0.75\,\alpha\,(1+Kn)}{Kn^2 + Kn + 0.283\,Kn\,\alpha + 0.75\,\alpha}$$    (2)

where, $\alpha$ is the HNO₃ mass accommodation coefficient, specified to be 0.2 (Jacob, 2000).

We evaluate the effect of p-NO₃ photolysis in SSA in the MBL by including the photolysis of NITs (yielding HONO and NO₂)

in our updated model. We note that NITs only corresponds to p-NO₃ associated with coarse-mode SSA. The current version

of the GEOS-Chem model also tracks p-NO₃ associated with accumulation-mode aerosols (identified as "NIT" in the model),

but does not track the fraction that is specific to accumulation-mode SSA. We therefore do not include photolysis of

accumulation-mode p-NO₃ in our analysis of MBL SSA nitrate photolysis. We discuss the implication of this model limitation

in Section 3.1. We also present results of a sensitivity simulation designed to explore the effect of accumulation-mode NIT

photolysis on continental boundary layer composition in Section 3.4.

Following Ye et al. (2016), the NITs photolysis rate coefficient ($J_{NITs}$) is scaled to the photolysis rate of HNO₃ ($J_{HNO_3}$). We

refer to this scaling coefficient as $J_{scale}$ in the rest of this paper. Box model analyses of measurements from NOMADSS, the

spring 2007 Reactive Halogens in the Marine Boundary Layer (RHaMBLe) project, and measurements made at Cape Verde

suggest $J_{scale}$ values of between 10-300 (Ye et al., 2016; Reed et al., 2017; Ye et al., 2017), with the variability possibly related

to differences in SSA composition at different locations in the MBL (Ye et al., 2017). In light of this uncertainty, we show





model results for a range of $J_{scale}$ values, and use measurements of $NO_x$ and HONO from CVAO to estimate reasonable bounds for this scaling factor. In the subsequent discussion, we use the notation $J_{scale}^n$, where the superscript $n$ indicates the value of $J_{scale}$. Note that in this notation, $J_{scale}^0$ refers to the 'no NITs photolysis' case. NITs photolysis is initially assumed to yield HONO and $NO_2$ in the molar ratio of 0.67:0.33 (Ye et al., 2016; Reed et al., 2017), but we also present results from a subset

of sensitivity simulations with the HONO:$NO_2$ molar yield set to 0.93:0.07 (Ye et al., 2017).

### 2.2 Measurements used for model evaluation

### 2.2.1 MBL NITs and HNO₃ measurements in the eastern Atlantic Ocean

During the RHaMBLe field campaign in May and June 2007, volatile water-soluble nitrate (dominated by and hereafter referred to as $HNO_3$) was sampled from the top of the CVAO tower over 2-hour intervals at nominal flow rates of 20 L min$^{-1}$

using tandem mist chambers (MCs), each of which contained 20 ml deionized water (Keene et al., 2009). To minimize artifact phase changes caused by mixing chemically distinct aerosol size fractions on bulk prefilters, air was sampled through a size-fractionating inlet that inertially removed super-μm-diameter aerosols from the sample stream. Sub-μm aerosol was removed downstream by an in-line 47-mm Teflon filter (Zefluor 2-μm pore diameter) that was changed daily. Samples were analyzed on site by ion chromatography (IC) usually within a few hours after recovery. Data were corrected based on dynamic handling

blanks that were loaded, briefly (few seconds) exposed to ambient air flow, recovered, processed, and analyzed using procedures identical to those for samples. Collection efficiencies were greater than 95 % and, consequently, corrections for inefficient sampling were not necessary. Relative precisions based on paired measurements varied as functions of concentration and typically averaged ±10 % to ±25 %. The average detection limit (DL; estimated following Keene et al. (1989)) was 12 pptv. Longer-term aerosol nitrate measurements were also made at CVAO, as part of a larger suite of aerosol measurements.

Samples were collected on quartz-fibre filters using high-volume sampler with a PM10 inlet, and were subsequently analysed by ion-chromotography. Here, we use the measurement summaries for summer marine air masses from 2007-2011 reported by Fomba et al. (2014).

During October and November 2003, the German research ship *Polarstern* cruised along a latitudinal transect through the

eastern Atlantic Ocean from Germany to South Africa. $HNO_3$ was measured from the ship at approximately 23 m above the water line over 2-hour intervals using the same MC-IC technique described above. The average DL was 2 pptv. During this cruise, ambient aerosols were also sampled in parallel over discrete daytime and nighttime intervals using a modified Graseby-Anderson Model 235 cascade impactor configured with a Liu-Pui type inlet, precleaned polycarbonate substrates, and quartz-fiber backup filters (Pallflex 2500 QAT-UP) (Keene et al., 2009). At an average sampling rate of 0.78 m$^3$ min$^{-1}$, the average

50 % aerodynamic cut diameters for the impaction stages were 20, 11, 4.3, 2.5, 1.2, and 0.65 μm. Bulk aerosol was sampled in parallel on quartz-fiber filters at an average flow rate of 1.1 m$^3$ min$^{-1}$. Impactors and bulk-filter cassettes were cleaned, dried, loaded, and unloaded in a Class 100 clean bench configured with impregnated filters on the inlet to remove alkaline-





and acidic-reactive trace gases. Exposed substrates and filters were halved, transferred to polypropylene tubes, sealed in glass mason jars, frozen, and express shipped from Cape Town to the Mount Washington Observatory, NH for analysis. Half sections of each substrate were extracted in 13 ml deionized water using a mini vortexer and sonication; half sections of exposed backup and bulk filters were similarly extracted in 40 ml deionized water. Substrate and filter extracts was analyzed

by IC for major ionic species including $NO_3^-$ and $Na^+$. The corresponding average DLs for bulk samples were 0.04 and 2.6 nmol m$^{-3}$, respectively; those for the GMD 0.46-µm size fraction were 0.05 and 3.5 nmol m$^{-3}$, respectively; and those for the larger individual size fractions were 0.01 and 0.1 nmol m$^{-3}$, respectively. Internal losses of super-µm aerosols within slotted cascade impactors of this type average about 25 % to 30 % (e.g. Willeke, 1975); other sources of bias for size-resolved particulate analytes based on the above procedures are generally unimportant (Keene et al., 1990).

**2.2.2 Long-term measurements of NO$_x$ and O$_3$ from CVAO**

Ambient air for NO$_x$ and O$_3$ measurements is drawn from a 40 mm glass manifold (QVF) with a hooded inlet positioned 10 m above ground level at the CVAO site. NO and NO$_2$ are measured by NO chemiluminescence (Drummond et al., 1985) with NO$_2$ converted photolytically to NO by illumination at 385-395 nm (Lee et al., 2009). The limits of detection for NO and NO$_2$ were 0.30 and 0.35 pptv when averaged over an hour, with an accuracy of 5 % and 5.9 % respectively. Reed et al., (2016)

showed that in regions that typically have low NO$_x$ concentrations and are cold, *in situ* measurements of NO$_2$ may be instrumentally biased high due to thermal decomposition of NO$_y$ species such as PAN and HO$_2$NO$_2$, leading to apparent perturbations in the Leighton relationship (Leighton 1961). These effects are however considered to be minimal at CVAO (Reed et al., 2016). Ozone (O$_3$) is measured by a Thermo Scientific model 49i UV photometer which quantifies O$_3$ by single wavelength absorption at 254 nm along a 38 cm path-length whilst simultaneously determining a zero background

measurement in a second absorption cell. The lower detectable limit is 1ppbv.

**2.2.3 Short-term measurement of HONO from CVAO and RHaMBle**

Measurements of the HONO at CVAO were performed between 24$^{th}$ November and 3$^{rd}$ December 2015 using a LOng Path Absorption Photometer (LOPAP-03 QUMA Elektronik & Analytik GmbH), described in Heland et al. (2001). Briefly, the LOPAP is a wet chemical technique where gas-phase HONO is sampled within a stripping coil into an acidic solution and is

derivatized into an azo dye, with the light absorption of the dye at 550 nm measured with a spectrometer. The LOPAP was operated and calibrated according to the standard procedures described in Kleffmann and Wiesen (2008) and was set up at CVAO in order to maximise the sensitivity of the LOPAP, as described in Reed et al. (2017). Regular baseline measurements (8 hrs) using an overflow of high purity N$_2$ were performed to account for instrument drift. The detection limit (2σ) of the LOPAP was calculated to be 0.2 pptv, with the relative error conservatively set to 10 % of the measured concentration.



During RHaMBLe (see Lee et al., 2010; Sander et al., 2013) HONO was measured in parallel with HNO$_3$ using the same MC-IC technique described above. The average detection limit was 2 pptv. These data are considered semi-quantitative because performance of the MC-IC technique for HONO has not been rigorously evaluated. However, intercomparisons of opportunity with HONO measured in parallel by MC-IC and long-path DOAS during summer 2004 in coastal New England as part of

ICARTT (ranging from <45 to 730 pptv; Keene et al., 2006, 2007) and by MC-IC and NI-PT-CIMS during winter 2011 in Colorado as part of NACHTT (ranging from <32 to 200 pptv; VandenBoer et al., 2013) indicate generally good agreement within ±20 %.

**2.3 Model simulations**

We performed a series of 2-year model simulations (each with a 6-month spin up from a standard GEOS-Chem benchmark

simulation) covering the 2014-2015 period, so as to compare with long-term measurements of NO$_x$ and O$_3$ at CVAO during this period. Given our focus on the MBL, we used the 4$^o$ x 5$^o$ horizontal resolution version of the model for computational efficiency. The vertical grid is a 47-level hybrid sigma pressure grid, extending from the surface to 80 km. The vertical grid resolution in the lowermost km is about 125-150 m. All simulations used meteorological fields from the Goddard Earth Observation System Forward Processing (GEOS-FP) assimilation system produced by the NASA Global Modeling and

Assimilation Office (Luchesi, 2013). Data processing was performed using the open-source python packages Numpy [van der Walt et al., 2011] and Pandas [McKinney, 2010], and figures were prepared using the open-source python plotting packages matplotlib [Hunter et al., 2007] and Seaborn [Waskom et al., 2017]

In the rest of this paper, we refer to the model with the imposed alkalinity limitation to acid uptake as the 'standard' model,

and the model with additional nitric acid uptake due to hydrochloric acid displacement (described in Section 2.1) as the 'updated' model.

**3. Results and discussion**

**3.1 Gas-particle partitioning of nitrate in the MBL**

Figure 1 shows a comparison of modeled gas-phase HNO$_3$ and total particulate nitrate (i.e. the sum of NITs and NIT)

concentrations with measurements in the tropical eastern North Atlantic for the standard model and for the updated model. We note that the modeled time period does not coincide exactly with the different periods over which the HNO$_3$ and p-NO$_3$ measurements were taken, indicated in Fig. 1. However, interannual variability in these regions is typically low (Carpenter et al., 2010), while the changes in HNO$_3$ and total nitrate concentrations induced by the model update are large. It is clear from Fig. 1 that the standard model does not capture the relative partitioning of nitrate between the gas- and particle- phases in the

MBL. In the standard model, most of the nitrate is in the gas-phase, in contrast to the measurements which show that most nitrate is in the particle phase.



The biased simulation of p-NO$_3$/HNO$_3$ ratios in the standard model is the result of the imposed alkalinity limitation to acid uptake (see section 2.1). For example, tropical North Atlantic SSA emission fluxes are of the order of 25-50 kg km$^{-2}$ day$^{-1}$ (Jaeglé et al., 2011) and the typical alkalinity of freshly emitted SSA is equal to 0.07 equivalents per kg (Gurciullo et al., 1999).

Assuming a 1 km MBL depth, this translates to maximum NITs production rates in the MBL of only 1.7-3.5 pptv hr$^{-1}$ in the standard model. In fact, actual NITs production rates in this scheme will be even lower because of competition with gas-phase sulfur dioxide uptake. These low production rates, coupled with the short lifetime of NITs against deposition, is the cause of the biased simulation of MBL p-NO$_3$/HNO$_3$ ratios in the standard model. In the updated model, HNO$_3$ uptake by coarse-mode SSA continues after alkalinity is titrated due in part to hydrochloric acid displacement (as described in Section 2.1) and the

simulation of the MBL nitrate partitioning between gas- and particle phases is much improved. This improvement in the simulation of gas-particle partitioning of nitrate is important in the context of our focus on assessing the large-scale importance of p-NO$_3$ photolysis on MBL chemistry. While this change improves the model performance, it is evident from Fig. 1 that the model underestimates p-NO$_3$ concentrations for all of the sites evaluated. Given that the dataset of simultaneous remote oceanic HNO$_3$ and p-NO$_3$ observations is small, it is unclear at this stage what could be causing this discrepancy.

We also note that simulated SSA concentrations are in agreement with measurements at CVAO. The Na$^+$ concentration measurements atop a 30 m tower (40 m above sea level) during RHaMBLE (Lawler et al., 2009) imply a SSA mass concentration of 6-19 µg m$^{-3}$. Fomba et al. (2014) reported similar SSA concentrations (11 ± 5.5 µg m$^{-3}$) at the top of the tower over a 5-year period from January 2007 to December 2011. In comparison, the simulated annual-average SSA concentration

at the lowest model level (centered ~60 m above sea level) at this location is 12.5 µg m$^{-3}$.

Figure 2 shows maps of average 2014-2015 simulated surface mixing ratios of HNO$_3$ and NITs in the MBL for the standard and updated models. Consistent with the discussion above, gas-phase HNO$_3$ is the dominant form of nitrate in the MBL in the standard model, with mixing ratios exceeding 100 pptv over ocean basins downwind of major continental NO$_x$ source regions.

By contrast, maximum simulated NITs mixing ratios are about 10-20 pptv over ocean basins. The updated acid uptake scheme changes the simulated partitioning of nitrate in the MBL globally. Other than in the immediate vicinity of continental NO$_x$ source regions, simulated nitrate is predominantly associated with the particle phase everywhere in the MBL in the updated model (see also Fig. S1). In the rest of this paper, we use the updated model to assess the effects of NITs photolysis on MBL oxidant chemistry.

As mentioned earlier, the current version of the GEOS-Chem model tracks accumulation-mode p-NO$_3$, but does not separately track the fraction of this mode that is associated with SSA. Since coarse-mode NITs is the dominant form of modeled p-NO$_3$ in the MBL in continental outflow regions where concentrations are highest (Figs. 2 and S1), this model limitation is not



particularly significant in terms of our analysis of SSA p-NO$_3$ photolysis. We plan to further evaluate this with a future version of the model that will explicitly track p-NO$_3$ in accumulation-mode SSA.

**3.2 NO$_x$ and HONO diurnal cycles at CVAO**

The diurnal variation of NO$_x$ and HONO in the MBL can potentially provide useful information on *in situ* NO$_x$ chemical

production and destruction. In the remote MBL, one would expect NO$_x$ concentrations to decrease over the course of the day in the absence of an *in situ* chemical source that peaks during the day, owing to the shorter lifetime against oxidation by OH during daylight hours. Similarly, HONO has a lifetime of the order of minutes against photolysis (e.g. Kraus and Hofzumahaus, 1998), and so a diurnal profile that peaks during the day would provide strong evidence for an *in situ* photochemical source for HONO in the MBL. Measurements at CVAO of NO$_x$ and HONO in air masses representative of relatively aged marine

boundary layer air do indeed show HONO and NO$_x$ peaking during the day (Reed et al., 2017). This fact has been used by Reed et al. (2017) and Ye et al. (2017) to argue for an *in situ* source of HONO and NO$_x$ from p-NO$_3$ photolysis based on box model analysis of measurements. Their analyses are, however, limited by the fact that background NO$_x$ levels cannot be explicitly simulated in a box modeling framework. To overcome this limitation, Reed et al. (2017) modeled the background source of MBL NO$_x$ from PAN decomposition by prescribing a flux of PAN from the free troposphere to the MBL. Ye et al.

(2017) constrained background NO$_x$ levels in their analysis by specifying NO$_2$ mixing ratios after sunrise. Here, we re-analyze the CVAO NO$_x$ and HONO measurements in the context of our three-dimensional model that explicitly simulates transport of NO$_x$ and its precursors.

Figures 3, 4, and 5 show comparisons of simulated diurnal profiles of NO, NO$_2$, and NO$_x$ with measurements at CVAO. Model

results are shown for $J_{scale}$ ranging from 0 (i.e. $J_{scale}^0$, no NITs photolysis) to 100 (i.e. $J_{scale}^{100}$). To mimic the fact that air masses arriving at CVAO are representative of aged background MBL air (Read et al., 2008; Carpenter et al., 2010), model results are shown for the grid box to the north-west of CVAO to minimize the effect of fresh NO$_x$ emissions from ship traffic off the coast of north Africa on model-measurement comparisons.

Several aspects of the comparisons shown in these figures are worth noting. In all seasons, except winter, a weak daytime peak is evident in observed NO$_x$. This peak is the result of a relatively flat NO$_2$ diurnal cycle, combined with an NO diurnal cycle that peaks strongly during the day. The model has difficulty reproducing the details of this diurnal variation. The shape of the NO diurnal cycle is generally well captured by the model, though peak model NO is generally higher than the measurements when $J_{scale}$ is larger than 25. The modeled NO$_2$ diurnal cycle consistently shows a daytime minimum in the

absence of NITs photolysis, a feature that is not evident in the measurements. Modeled NO$_2$ agrees better with measurements when $J_{scale}$ is specified to be 25-50, though the exact shape of the NO$_2$ diurnal cycle is not well simulated. This is, in part, due to the sharp decrease in simulated NO$_2$ at sunrise, owing to the formation and hydrolysis of halogen nitrates. The absence of this feature in the measurements has been previous noted by Reed et al. (2017). They speculate that this could be because of



missing halogen chemistry during night-time and/or during the night/day transition in the model. While there are likely shortcomings in our model's representation of NO/NO$_2$ chemistry, the diurnal NO$_x$ comparisons (Fig. 5) provide support for an *in situ* source of NO$_x$ from NITs photolysis, with NITs photolysis coefficients about 25-50 times larger than corresponding HNO$_3$ photolysis coefficients. We compare our estimates of the NITs photolysis coefficient with the corresponding estimates

by Reed et al. (2017) and Ye et al. (2017) at the end of this sub-section.

Figure 6 shows comparisons of the modeled HONO diurnal cycle with short-term measurements from the spring 2007 RHamBle campaign and with the November-December 2015 CVAO measurements. The measurements consistently show a strong daytime peak in HONO, with peak median mixing ratios ranging from 3.5 to 5.5 pptv. The model reproduces the

observed shape of the HONO profile only when NITs photolysis in included. Further, simulated HONO mixing ratios are less than 0.1 pptv when NITs photolysis is not considered, and the model underestimates peak HONO even for $J_{scale}^{100}$. It is worth recognizing, however, that simulated HONO mixing ratios at CVAO are roughly proportional to the assumed molar yield (set at 0.67) of HONO from NITs photolysis. This is illustrated by the dashed lines in Fig. 6, showing simulated HONO from $J_{scale}^{50}$ and $J_{scale}^{100}$ sensitivity runs in which the molar yield of HONO from NITs photolysis is increased to 0.93 (with a concomitant

reduction of the NO$_2$ yield to 0.07). The choice of 0.93 for the HONO yield in the sensitivity runs is based on the budget analysis of the RHaMBLe measurements by Ye et al. (2017). Increasing the HONO yield from 0.67 to 0.93 increases peak simulated HONO mixing ratios by about 35 %, bringing them into better agreement with the measurements. We note that changing the relative yields of HONO and NO$_x$ from NITs photolysis does not significantly affect simulated NO$_x$ levels because of the short lifetime of HONO against photolysis to NO.

While our analysis supports the existence of an *in situ* photolytic source of HONO and NO$_x$ in the MBL, we are unable to draw a definitive conclusion on the magnitude of this source. The NO$_x$ comparisons presented here suggests that $J_{scale}$ ranges from 25-50, while the HONO comparisons support a larger value of $J_{scale}$ and/or HONO yields from NITs photolysis being close to unity. In the context of assessing the large-scale impact of NITs photolysis, we therefore conclude that a range of 25-100 is

the most appropriate range for the parameter $J_{scale}$ in our model. We note that our best estimates of $J_{scale}$ are about a factor of 4-5 higher than the corresponding scaling coefficient estimated by Reed et al. (2017) and Ye et al. (2017) based on their box model analysis. This is because simulated NITs mixing ratios at CVAO in our model (median of ~100 pptv) are about 4 times lower than the p-NO$_3$ mixing ratios (~300-400 pptv) prescribed by Reed et al. (2017) and Ye et al. (2017) in their box models. As noted in Section 3.1, further work is needed to resolve this model discrepancy.

**3.3 Impact of NITs photolysis on global NO$_x$, OH, and O$_3$**

Figure 7 shows the simulated MBL-average distribution of NO$_x$, OH, and O$_3$ from the $J_{scale}^0$ model run, as well as the relative enhancements (relative to the $J_{scale}^0$ case) of simulated mixing ratios for the $J_{scale}^{25}$ and $J_{scale}^{100}$ model runs. In the $J_{scale}^0$



simulation, highest MBL mixing ratios of both $NO_x$ and $O_3$ occur downwind of mid-latitude Northern Hemisphere continental source regions, and, to a lesser spatial extent, downwind of tropical biomass burning regions for $O_3$. By contrast, highest OH concentrations over oceans occur between 30°S-30°N, reflecting the latitudinal pattern of primary OH production from $O_3$ photolysis.

As expected, the additional source of $NO_x$ from NITs photolysis in the $J_{scale}^{25}$ and $J_{scale}^{100}$ model runs leads to enhancements in simulated $NO_x$, OH, and $O_3$ concentrations in the MBL. Downwind of continental regions, MBL $NO_x$ mixing ratios increase by 5-50 pptv in the $J_{scale}^{25}$ simulation, and by 10-100 pptv in the $J_{scale}^{100}$ simulation, relative to the $J_{scale}^{0}$ simulation (Fig. S2). MBL OH mixing ratios increase by $10^5$-$10^6$ molecule cm$^{-3}$, and MBL $O_3$ mixing ratio increase by 1-5 ppbv over these same

regions (Fig. S2). The simulated relative impact of NITs photolysis peaks over the tropical and subtropical oceans, with enhancement factors ranging from 5-20 for $NO_x$, 1.2-1.6 for OH, and 1.1-1.3 for $O_3$ (Fig. 7). The spatial pattern of these relative enhancements due to NITs photolysis is thus very different from the spatial distribution of NITs itself (see Fig. 2). The importance of the tropics is due to intense solar radiation and high temperatures. These conditions lead to an enhanced source of $NO_x$ from nitrate photolysis, together with a reduced source from organic nitrates which have a short lifetime in the warmer

air and so have lower concentrations over the remote oceans. This can be understood by comparing the spatial pattern of the production rate of $NO_x$ from NITs photolysis and PAN decomposition in the $J_{scale}^{25}$ and $J_{scale}^{100}$ model runs with the spatial pattern of the production rate of $NO_x$ from PAN decomposition in the $J_{scale}^{0}$ model run, as shown in Fig. 8. In the MBL at Northern Hemisphere mid-latitudes, the chemical source of $NO_x$ from PAN decomposition in the $J_{scale}^{0}$ simulation is generally much larger than the source from NITs photolysis in the $J_{scale}^{25}$ simulation. This is also true for the upper-limit $J_{scale}^{100}$ simulation,

though to a lesser extent. By contrast, the $NO_x$ source from NITs photolysis is significantly larger than the $NO_x$ source from PAN decomposition over most of the tropical MBL. Inclusion of NITs photolysis in the model therefore more significantly impacts $NO_x$ chemical production, and consequently, simulated $NO_x$ mixing ratios, in the tropical MBL. The impact of this additional $NO_x$ source from NITs photolysis on MBL $NO_x$ mixing ratios is modulated by the reduction in $NO_x$ lifetime due to the increase in model OH when NITs photolysis is included (see Fig. 7). We also note that the inclusion of NITs photolysis

improves the simulation of $O_3$ at CVAO (Fig. 9), offsetting to some extent the simulated effect of halogen-mediated chemical loss of $O_3$ (Sherwen et al., 2016b) in the MBL.

While the inclusion of NITs photolysis is important in terms of modeled $NO_x$, OH, and $O_3$ in the tropical marine atmosphere, it is important to note that its significance is limited to the boundary layer owing to the short lifetime of coarse-mode SSA, and

thus of NITs, against deposition. This is illustrated in Fig. 10, which shows simulated enhancements in $NO_x$, $O_3$, and OH due to NITs photolysis for a vertical cross-section at 20° W. In all cases, the impact of NITs photolysis on $NO_x$, OH, and $O_3$ is significant only for altitudes below 1.5-2 km. Very little coarse-mode SSA gets out of the boundary layer and so the impact of NITs photolysis is limited to essentially the MBL. This limits the global impact of this new process. Although Fig. 9 shows



some large changes in MBL concentrations, when averaged over the whole troposphere these changes are relatively small. Relative to the $J_{scale}^0$ model run, increases in the tropospheric masses of NO$_x$, OH, and O$_3$ in the $J_{scale}^{25}$ ($J_{scale}^{100}$) model runs are 0.6 % (1.7 %), 1.1% (2.9 %), and 0.5% (1.4 %), respectively.

### 3.4 Impact of accumulation-mode p-NO$_3$ photolysis in continental regions

Although much of the recent research emphasis has been on the photolysis of nitrate in SSA, there is some evidence that this might be a more general phenomenon and that p-NO$_3$ photolysis may also occur in other aerosol types. For example, Ye et al. (2018) found p-NO$_3$ photolysis to be a significant source of HONO in background terrestrial air masses over the southern United States. In order to illustrate the potential large-scale impact of this process, we performed one additional simulation in which we allow for photolysis of both NITs and accumulation-mode nitrate (identified as "NIT" in the model). This

accumulation-mode aerosol, composed of sulfate, ammonium, and nitrate, is the dominant form of p-NO$_3$ over continental regions. In this simulation, we set the photolysis rate scaling coefficient (i.e. $J_{scale}$) to be 25 (chosen for illustrative purposes) for both NIT and NITs, and assume that the HONO:NO$_2$ molar yield is 0.67:0.33 as in the $J_{scale}^{25}$ simulation. We refer to this simulation by the notation $J_{scale}^{25*}$. The effect of NIT photolysis is isolated by comparing the $J_{scale}^{25*}$ simulation with the $J_{scale}^{25}$ simulation.

Figure 11 shows simulated annual-average NIT and NO$_x$ mixing ratios in the boundary layer from the $J_{scale}^{25}$ simulation, and the change in these mixing ratios when NIT photolysis is included in the $J_{scale}^{25*}$ simulation. Simulated NIT and NO$_x$ mixing ratios are highest (> 1 ppbv) in the eastern United States, western Europe, eastern China, and northern India. Adding NIT photolysis decreases NIT mixing ratios by around 0.01-0.1 ppbv in these regions. Surprisingly, the expected increase in NO$_x$

from NIT photolysis is seen in the model results only in northern India. In fact, adding NIT photolysis decreases simulated NO$_x$ in the eastern United States, western Europe, and most notably in eastern China. There are also a few model grid boxes, especially in southern India and eastern China, where NIT mixing ratios increase when NIT photolysis is included. These counterintuitive features are due to the fact that p-NO$_3$ photolysis is also a source of OH, and NIT photolysis increases OH in all continental source regions (Fig. 12). The net impact of NIT photolysis on NO$_x$ in a region is determined by the balance

between the relative importance of the increased NO$_x$ source due to NIT photolysis and the decreased NO$_x$ lifetime due to the increase in OH. The balance between these two counteracting effects on NO$_x$ varies from region to region in the model. In all cases though, adding NIT photolysis leads to an increase of 1-2 ppbv in O$_3$ in most regions, with larger increases (>6 ppbv) seen in northern India and eastern China. From a tropospheric average perspective, the impact of NIT photolysis is significantly larger than the impact of NITs photolysis. Relative to the $J_{scale}^{25}$ model run, increases in the tropospheric masses of NO$_x$, OH,

and O$_3$ in the $J_{scale}^{25*}$ model run are 2.8 %, 4.6 %, and 2.6 %, respectively.





Our results highlight the potential importance of p-NO$_3$ photolysis in continental high-NO$_x$ source regions. We note, however, that the coarse model resolution used in this study cannot resolve important chemical gradients in continental source regions. If further field studies convincingly demonstrate the occurrence of p-NO$_3$ photolysis on aerosols other than SSA, finer resolution regional model simulations will be needed to more accurately assess the effect of p-NO$_3$ photolysis on continental

boundary layer oxidant chemistry.

### 4. Conclusions

We have presented the first global perspective of the impact of NITs photolysis on MBL oxidation chemistry. Comparisons with measurements at CVAO shows that this mechanism is important in terms of explaining some observed aspects of MBL oxidant chemistry. Specifically, we find that the diurnal variation of both HONO and NO$_x$ at CVAO, as well as the absolute

magnitude of HONO mixing ratios, are consistent with a NITs photolysis coefficient that is about 25-100 times larger than the corresponding gas-phase HNO$_3$ photolysis coefficient. Relative to a base case simulation with no NITs photolysis, simulations with NITs photolysis coefficients of this magnitude result in peak relative enhancements ranging from factors of 5-20 for NO$_x$, 1.2-1.6 for OH, and 1.1-1.3 for O$_3$ in the tropical and subtropical marine boundary layer. We have also presented preliminary analysis showing the potential importance of p-NO$_3$ photolysis for continental boundary layer oxidant chemistry.

Our results point to the need for extending and enhancing the MBL NO$_x$ chemistry measurements at CVAO and elsewhere. In particular, accurate measurements of the diurnal variation of HONO at a variety of MBL locations can provide further insight into the impact of NITs photolysis on reactive nitrogen cycling over oceanic regions. From a mechanistic perspective, laboratory and field measurements are needed to better understand the mechanism of ambient p-NO$_3$ photolysis, including the

extent to which nitrate photolysis may occur on other aerosol types over both land and ocean regions. Further model developments are also needed to delineate the fraction of nitrate associated with fine-mode sea-salt aerosol and to better couple the calculation of acid uptake on sea-salt aerosols with the MBL halogen chemistry mechanism used in the model. High resolution regional model studies are needed to better understand impacts of nitrate photolysis in continental source regions. Future model studies will be needed to assess the implications of p-NO$_3$ photolysis with regards to interpreting oxygen isotopic

composition measurements of nitrate (Alexander et al., 2009; Savarino et al., 2013) and assessing changes in the tropospheric O$_3$ distribution from pre-industrial to present.

### 5. Data availability

Request for the specific version of the model code and model output should be addressed to Prasad Kasibhatla (psk9@duke.edu). The statistical measurement summaries used in Figure 1 are available in the papers referenced in the figure



caption. Measurements from CVAO (Figures 3-6, 9) are available from the Centre for Environmental Data Analysis (CEDA, http://www.ceda.ac.uk/).

**Acknowledgements**

TS, MJE, LJC, CR, JDL, KAR acknowledge the financial support of the National Centre for Atmospheric Science (NCAS) for maintaining CVAO and the Natural Environmental Research Council (NERC) though grants NE/L01291X/1, NE/E011330/1, and NE/D006554/1. The 2015 HONO measurements at CVAO were support by the NERC project SNAABL, NE/M013545/1. BA acknowledges funding from the National Science Foundation through award AGS-1446904. Financial support for the University of Virginia was provided by the National Science Foundation through awards AGS-0328342 and

AGS-0646854. Financial support for the University of New Hampshire was provided by the National Science Foundation through awards AGS-0328389 and AGS-0646864.

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





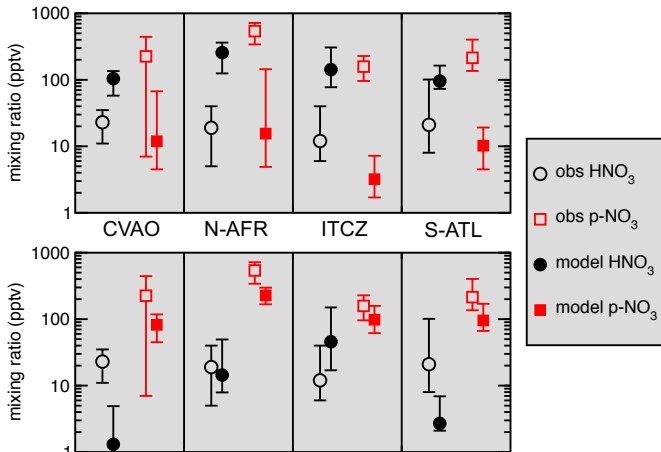

Figure 1: Comparison of modeled gas-phase HNO₃ and accumulation- plus coarse-mode particulate nitrate (p-NO₃) from the standard (top panel) and updated (bottom panel) models with measurements. CVAO HNO₃ measurements are for open ocean air masses from the spring 2007 RHaMBLe campaign (Lawler et al., 2009), while p-NO₃ measurements are from the summer marine aerosol subset of samples from 2007-2011 (Fomba et al., 2014). N-AFR, ITCZ, and S-ATL HNO₃ and p-NO₃ ship-board measurements are for the North Africa, ITCZ, and South Atlantic transport regimes in the eastern Atlantic during November 2003 (Keene et al., 2009). Model results are for May (CVAO HNO₃), June-August (CVAO p-NO₃), and November (N-AFR, ITCZ, and S-ATL HNO₃ and p-NO₃) from the 2014-2015 simulations. Symbols and lines indicate means and ranges and/or 2 standard deviations for measurements, and medians and 5th and 95th percentiles of daily means for models.





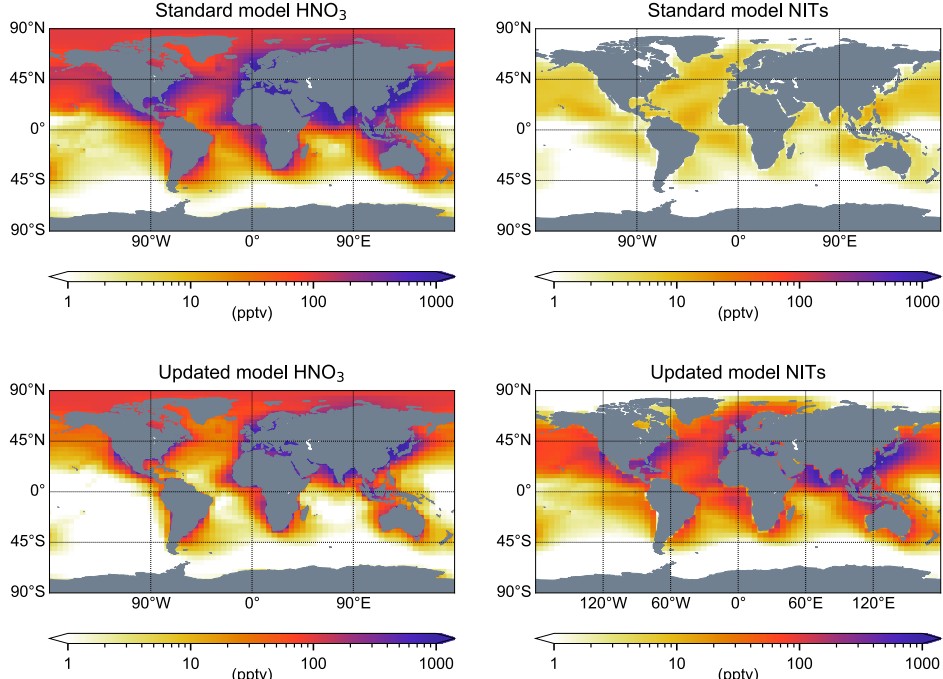

Figure 2: Simulated 2014-2015 MBL-average (average over bottom 1 km) $HNO_3$ and NITs (p-$NO_3$ associated with coarse-mode SSA) for the standard and updated models.





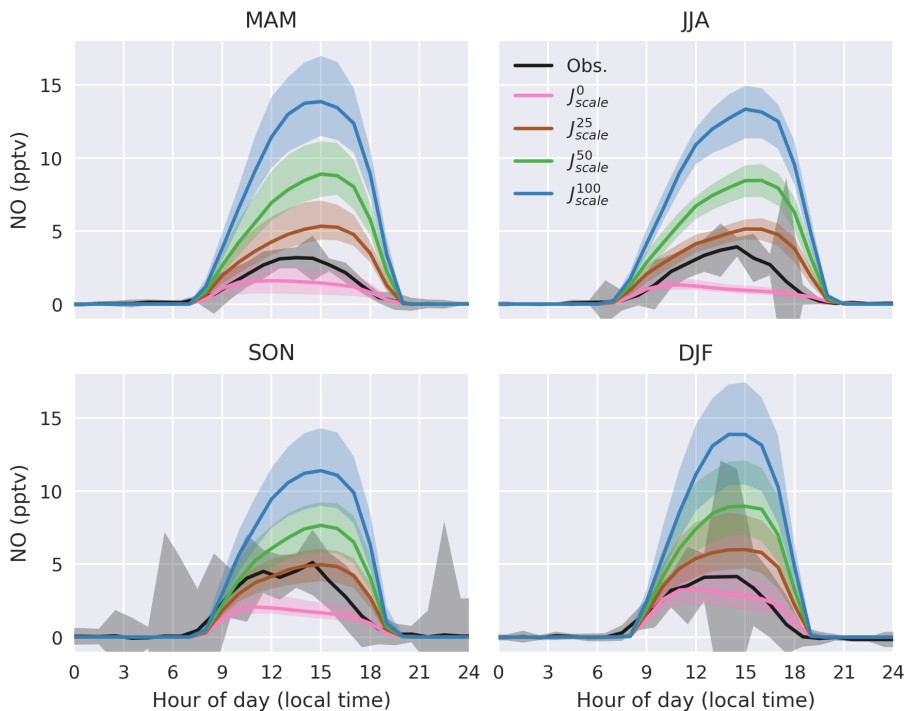

Figure 3: Observed and modeled diurnal cycles of NO at CVAO during different seasons (MAM: March, April, May; JJA: June, July, August; SON: September, October, November; DJF: December, January, February) in 2014-2015. Solid lines represent median values, and shaded regions show $1^{st}$ and $3^{rd}$ quartiles of model values and standard error of observations. Model simulations assume a HONO:NO$_2$ molar yield of 0.67:0.33 from NITs photolysis.





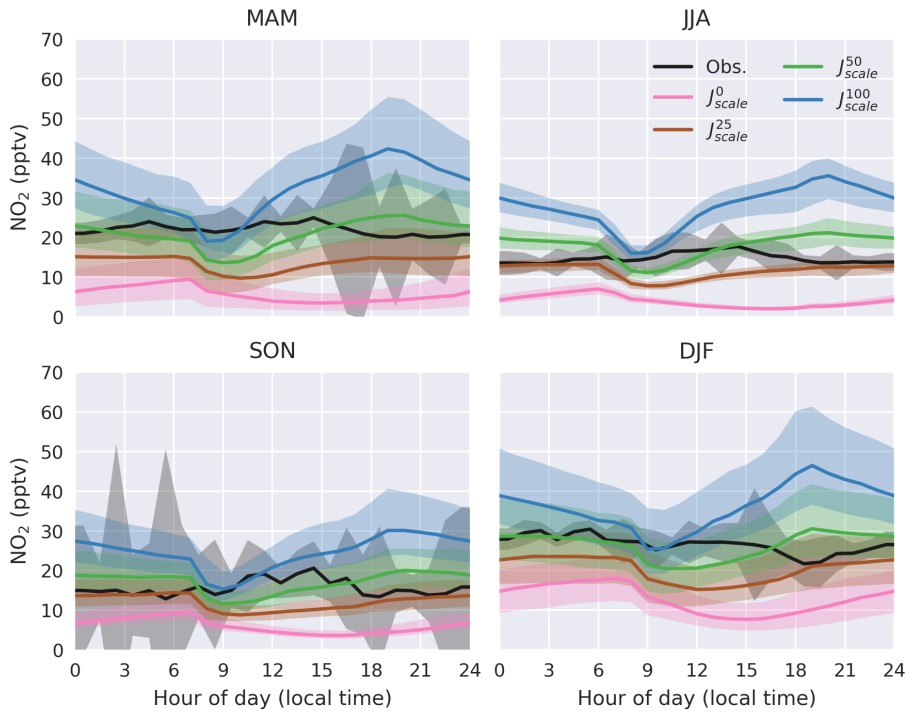

Figure 4: Observed and modeled diurnal cycles of NO$_2$ at CVAO during different seasons (MAM: March, April, May; JJA: June, July, August; SON: September, October, November; DJF: December, January, February) in 2014-2015. Solid lines represent median values, and shaded regions show 1$^{st}$ and 3$^{rd}$ quartiles of model values and standard error of observations. Model simulations assume a HONO:NO$_2$ molar yield of 0.67:0.33 from NITs photolysis.



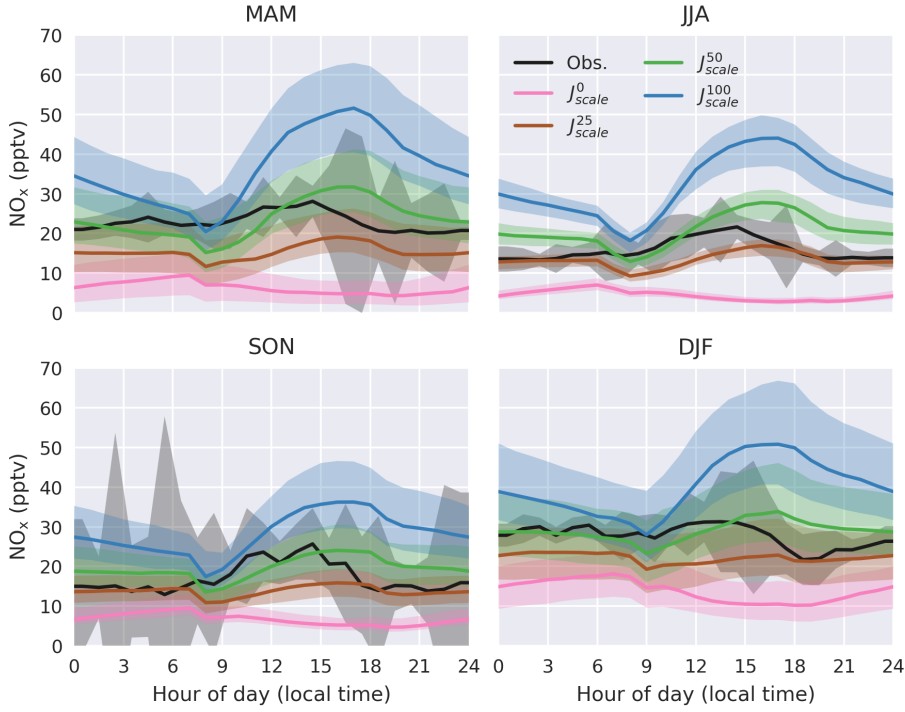

Figure 5: Observed and modeled diurnal cycles of $NO_x$ at CVAO during different seasons (MAM: March, April, May; JJA: June, July,
10 August; SON: September, October, November; DJF: December, January, February) in 2014-2015. Solid lines represent median values, and
shaded regions show 1st and 3rd quartiles of model values and standard error of observations. Model simulations assume a HONO:$NO_2$ molar
yield of 0.67:0.33 from NITs photolysis.





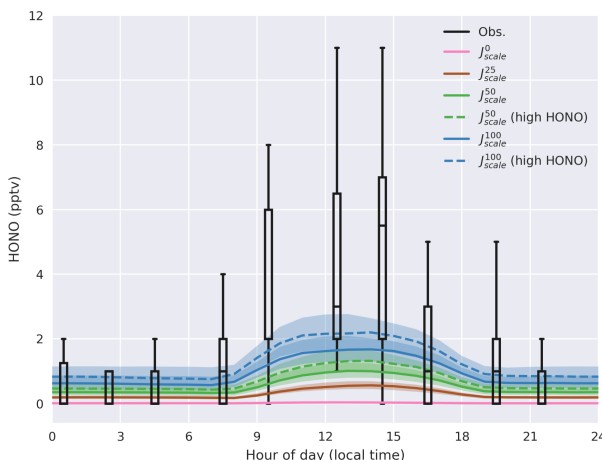

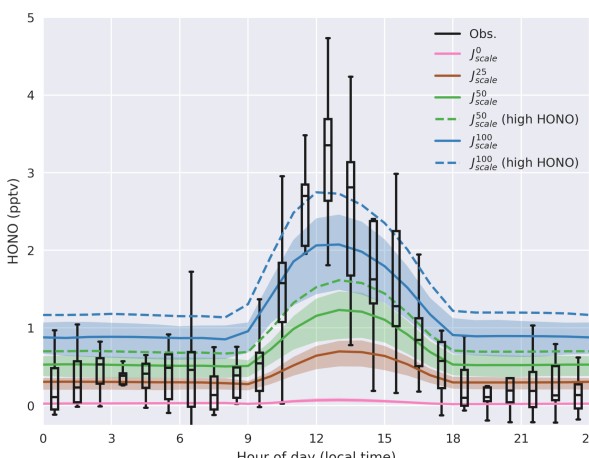

Figure 6: Observed and modeled diurnal cycles of HONO at CVAO. Top panel shown comparison of model results for May from the 2014-2015 simulations with measurements during the spring 2007 RHaMBLe campaign (Sander et al., 2013). Bottom panel shows comparison of model results with measurements (Reed et al., 2017) during November-December 2015. Measurements are shown in box-and-whisker notation, with the box showing the lower and upper quartile values (with a horizontal line at the median) and the whiskers extending to 1.5 times the interquartile range of the data. Solid and dashed lines represent model median values for simulations with a HONO:NO$_2$ molar yield of 0.67:0.33 and 0.93:0.07 (denoted as 'high HONO'), respectively, from NITs photolysis. Shaded regions show 1$^{st}$ and 3$^{rd}$ quartiles of model values.



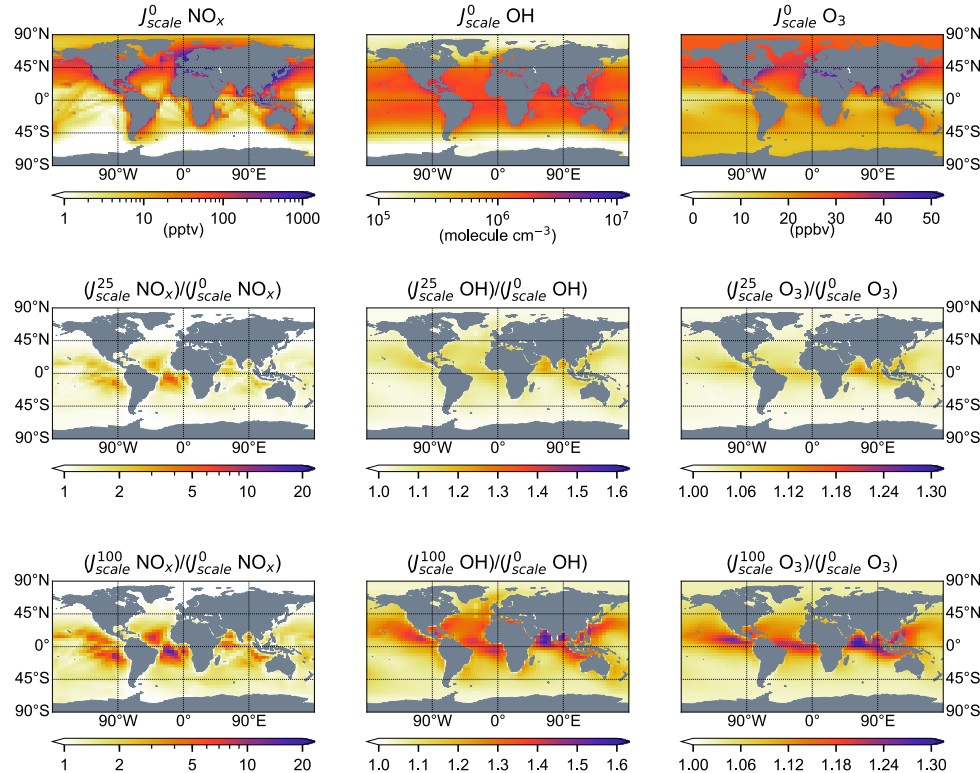

Figure 7: Simulated 2014-2015 MBL-average (average over bottom 1 km) concentrations for the $J_{scale}^0$ model run (top panels) and ratio of simulated MBL-average concentrations for the $J_{scale}^{25}$ (middle panels) and $J_{scale}^{100}$ (bottom panels) model runs relative to the $J_{scale}^0$ model run for $NO_x$ (left panels), OH (middle panels) and $O_3$ (right panels).



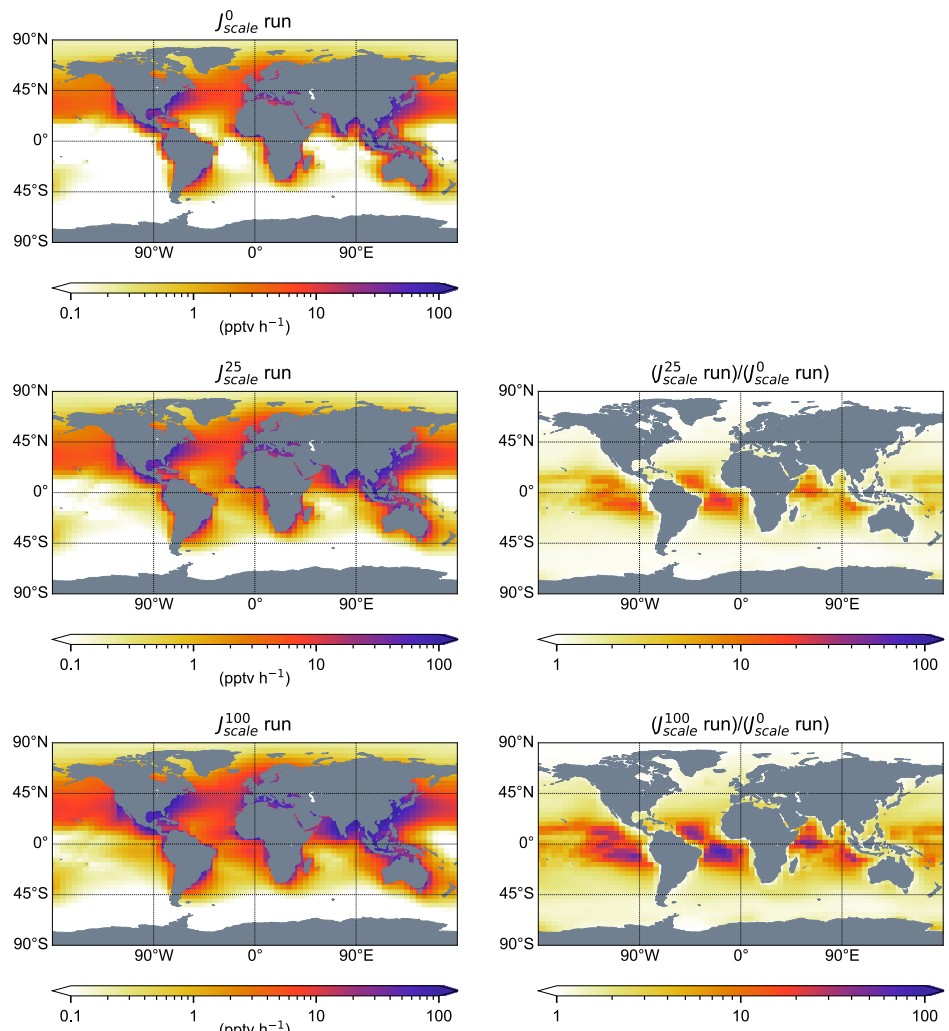

Figure 8: Simulated 2014-2015 MBL-average (average over bottom 1 km) $NO_x$ production rate from PAN decomposition for the $J_{scale}^0$ model run (top left panel) and from PAN decomposition and NITs photolysis for the $J_{scale}^{25}$ (middle left panel) and $J_{scale}^{100}$ (bottom left panel) model runs. The ratio of the $NO_x$ production rate from PAN decomposition and NITs photolysis for the $J_{scale}^{25}$ and $J_{scale}^{100}$ model runs relative to the $NO_x$ production rate from PAN decomposition for the $J_{scale}^0$ model run is shown in the middle right and bottom right panels, respectively.



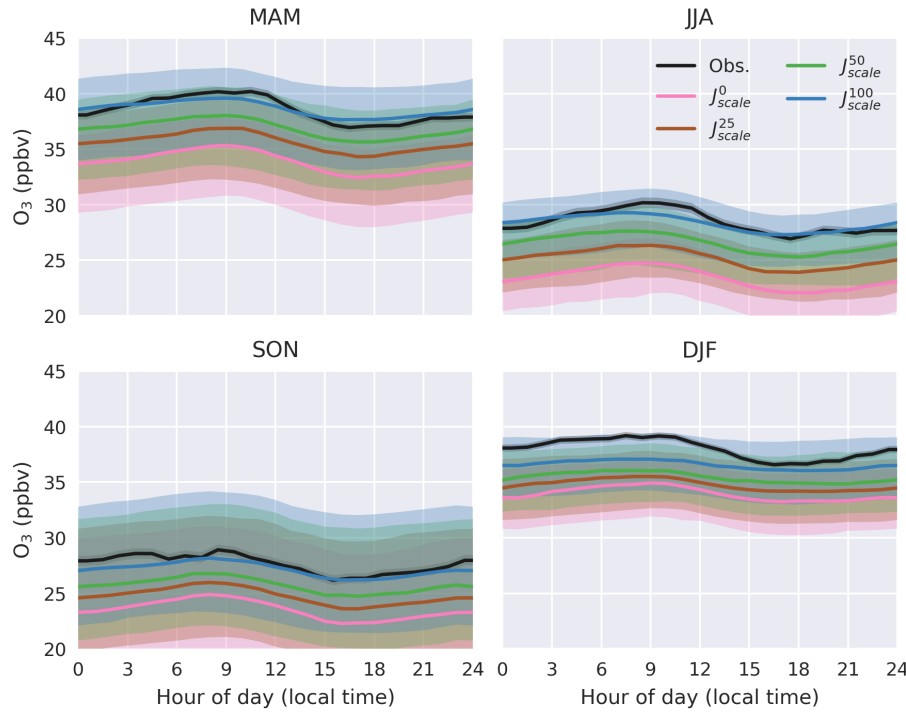

Figure 9: Observed and modeled seasonal cycle of $O_3$ at CVAO during 2014-2015. Solid lines represent median values, and shaded regions
10    show 1st and 3rd quartiles of model values and standard error of observations.



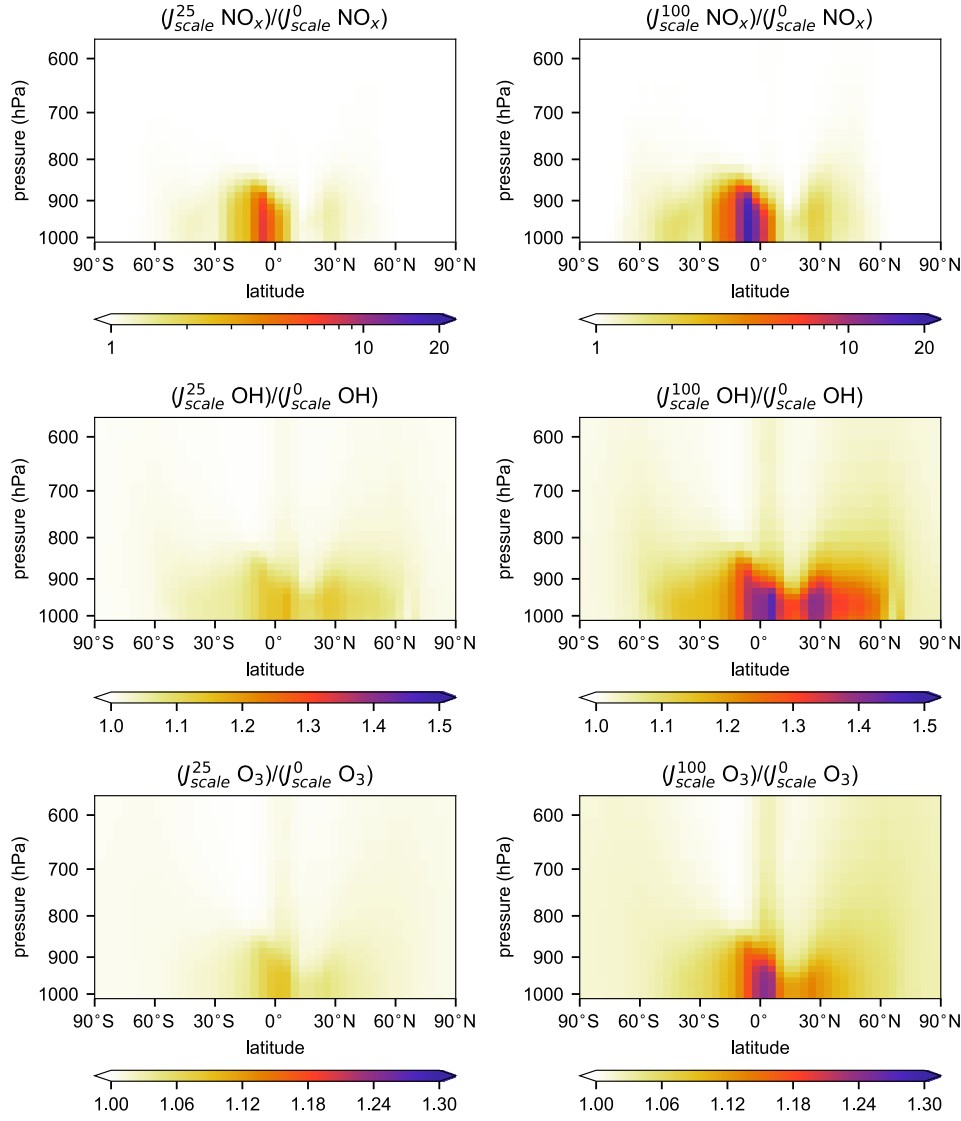

Figure 10: Latitude-altitude plot of average 2014-2015 $NO_x$ (top panels), OH (middle panels), and $O_3$ (bottom panels) for the $J_{scale}^{25}$ and $J_{scale}^{100}$ model runs (left and right panels, respectively) relative to corresponding fields for the $J_{scale}^{0}$ model run at 20° W longitude.



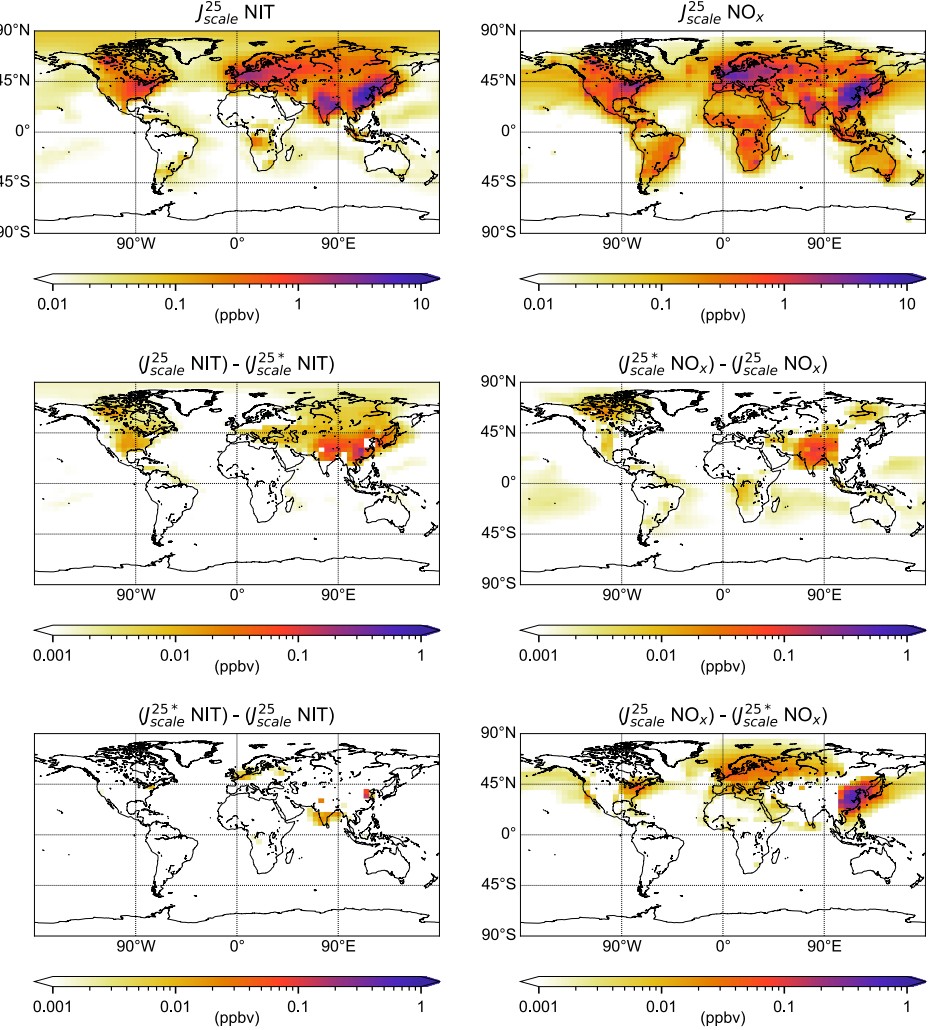

Figure 11: Simulated 2014-2015 boundary layer-average (average over bottom 1 km) accumulation-mode nitrate (NIT) and $NO_x$ mixing ratios for the $J^{25}_{scale}$ model run (top left and top right panels, respectively), and changes in NIT (middle left and bottom left panels) and $NO_x$ mixing ratios (middle right and bottom right panels) when NIT photolysis is included. Middle panels show regions where NIT ($NO_x$) mixing ratios decrease (increase) when NIT photolysis is included, and bottom panels show regions where NIT ($NO_x$) mixing ratios increase (decrease) when NIT photolysis is included.



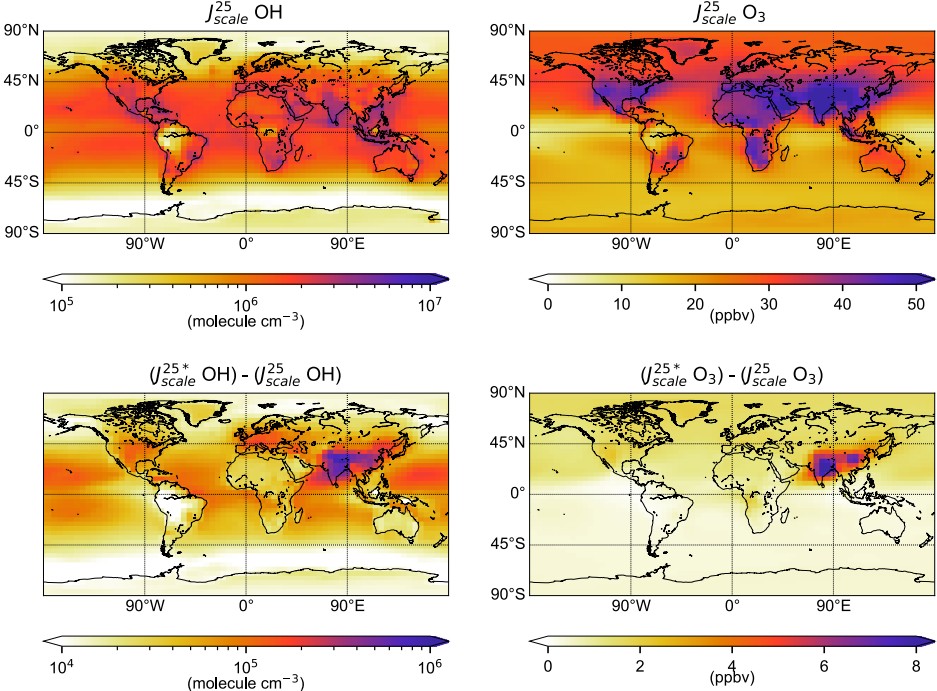

Figure 12: Simulated 2014-2015 boundary layer-average (average over bottom 1 km) OH and $O_3$ mixing ratios for the $J^{25}_{scale}$ model run (top left and top right panels, respectively), and changes in OH and $O_3$ mixing ratios ratios (bottom left and bottom right panels, respectively) when NIT photolysis is included.

