# Peer review of "Global impact of nitrate photolysis in sea-salt aerosol on $NO_x$ , OH, and $O_3$ in the marine boundary layer"

_Atmospheric Chemistry and Physics, 2018_

## Referee Comment (RC1) · Anonymous Referee #1 · 6 Jun 2018

In this manuscript, the authors have developed a sea-salt aerosol particulate nitrate photolysis module and incorporate it into the GEOS-Chem global model to evaluate the impact of this photochemical process on NOx, OH and O3 in the marine boundary layer (MBL). The modeling results indicate that while the impacts on global NOx, ozone and OH mass burdens are small (∼1-3 %), it could be significant on a regional scale in the tropical and subtropical marine boundary layer, with peak local enhancements ranging from factors of 5-20 for NOx, 1.2-1.6 for OH, and 1.1-1.3 for O3. The parameterizations appear to be appropriate for the stage of our current understanding (see below for the specific comments below). The manuscript is well organized and written, and is suitable for publication in Atmospheric Chemistry and Physics.

[Figure]

Specific comments:

1. The model appears to capture most of the variation features and the levels of NOx, HONO and particulate nitrate measured at the Cape Verde Atmospheric Observatory (CVAO) when appropriate ratios of NITs:HNO3 photolysis coefficients are used. However, it should be cautioned that the photolysis rate constant can be highly variable, by 2-3 orders of magnitude, for inorganic nitrate associated with aerosol particles in different regions (Ye et al., 2016, 2017a). Aerosol acidity and organic matrix are probably the two major factors in determining the NITs photolysis rate constant. While the authors have attempted to examine the effect of varying NITs photolysis rate constant by using different J(scale) values in the model, the validation through comparison with measurements from only one location at the CVAO is not sufficient. The authors should state explicitly the limitation of this modeling effort and the uncertainty in the global scale extrapolation. To truly understand the global impact of nitrate photolysis in sea-salt aerosol on NOx, OH, and O3 in the MBL, much more field studies in different marine environments are needed.

2. Is there a depositional term in the model for HONO, O3 and NO2? The authors did not give any explicit description on deposition parameterization; the answer to the question is probably yes, judging from significant diurnal variations in the modeled HONO concentration. Seawater is a slightly basic solution and is an effective sink for HONO and O3. Since measurements at the Cape Verde Atmospheric Observatory were made only a few meters above the sea level, the depositional loss can significantly affect the observed concentrations near the sea surface, for example, resulting in strong diurnal variations. The depositional loss may explain in part the lower HONO concentrations (Reed et al., 2017; Ye et al., 2017b) at the CVAO than those measured on board the C-130 aircraft (100 – 1000 m above sea level) (Ye et al., 2016). The depositional loss mechanism should be included in the model if it has not been yet.

References

Reed, C., Evans, M. J., Crilley, L. R., Bloss, W. J., Sherwen, T., Read, K. A., Lee, J. D., and Carpenter, L. J.: Evidence for renoxification in the tropical marine boundary layer, Atmos. Chem. Phys., 17, 4081–4092, doi:10.5194/acp-17-4081-2017, 2017.

Ye, C., Zhou, X., Pu, D., et al.: Rapid cycling of reactive nitrogen in the marine boundary layer, Nature, 532, 489–491, doi:10.1038/nature17195, 2016.

Ye, C., Zhang, N., Gao, H., and Zhou, X.: Photolysis of particulate nitrate as a source of HONO and NOx, Environ. Sci. Technol., 51, 6849−6856, DOI: 10.1021/acs.est.7b00387, 2017a.

Ye, C., Heard, D. E., and Whalley, L. K.: Evaluation of novel routes for NOx formation in remote regions, Environ. Sci. Technol., 51, 7442–7449, doi:10.1021/acs.est.6b06441, 2017b.

---

## Referee Comment (RC2) · Anonymous Referee #2 · 2 Jul 2018

This is a very nicely written paper that addresses some open questions regarding the role of particle nitrate photolysis in 'renoxification' using the GEOS-Chem model and several observational datasets. The measurements from CVAO, in particular, have been used in previous recent studies to explore this topic, but only using a box model, and the 3D model used here has the advantage of not needing to prescribe the impacts of transport. The portion of the analysis that explores the impact of removing an alkalinity limit of the conversion of HNO3 to particle phase nitrate on sea spray aerosol is important and likely to be of general interest. The model is also able to provide valuable information about the spatial extent of the impact of this sea salt-associated nitrate photolysis.

I recommend publication after the authors address the following comments: Page 3 Lines 25-30 The authors indicate that sea spray aerosol is generated/transported in two size fractions, and then go on to say that only the nitrate and sulfate associated with 'coarse mode' particles is designated as sea spray SO4S and NITS. Why not also explicitly track this sulfate and nitrate in the smaller particles? Is there some particular reason of is it related the history of the model? What is the consequence of excluding it for this analysis? This comes up again a couple of times and it might be useful to provide a bit more information here.

Page 4 Lines 29-32 What is the rationale for using JHNO3 to scale JNIT? Given that the rate of photolysis is so different, isn't it also likely that the wavelength dependence is also different? In that case, the scaling factor is likely to change with solar zenith angle. Was this considered?

Page 6, Lines 21-29 Was any particle filter used upstream of the LOPAP? If not, is it possible that some of the signal attributed to gas phase HONO might actually be related to particle phase nitrite associated with SSA? An artefact like this could lead to a strong positive bias in the inferred gas phase HONO measurements, with important implications for the radical budgets. In particular, it could explain why a scaling factor of 25-50 is sufficient to reproduce the NOx measurements at CVAO, but not the 'HONO' measurements.

Section 3.3 and Figure 8 Does the absolute rate of NOx production from PAN decomposition also change when nitrate photolysis is included? How much of the change in the NOx production rate is via this indirect impact as opposed to the nitrate photolysis itself? This may be a minor point, but would be interesting to know.

Figure 10 and last paragraph of section 3.3 – would this conclusion of the limited spatial extent of the impacts still be true if the nitrate in the accumulation mode SSA had been considered? It might be worth a sentence or two addressing this here. This is a slightly different question than what is being addressed in Section 3.4.

Section 3.4 The caption for Figure 11 and the associated text are very confusing. What do the asterisks denote – the inclusion of the photolysis scaling factor for all nitrate? That's not clear at all. I can understand why the authors are keen to extend the modelling exercise to probe the impact on nitrate photolysis more generally, but given the strongly non-linear results and the coarse model resolution, to me this section detracts from the rest of the manuscript rather than adding to it.

---

## Author Comment (AC1) · 16 Jul 2018

**Response to Reviewers**

We thank both reviewers for their thoughtful comments on the manuscript. We address below the specific comments of each reviewer.

**REVIEWER 1**

Reviewer 1, comment 1:
The model appears to capture most of the variation features and the levels of NOx, HONO and particulate nitrate measured at the Cape Verde Atmospheric Observatory (CVAO) when appropriate ratios of NITs:HNO3 photolysis coefficients are used. However, it should be cautioned that the photolysis rate constant can be highly variable, by 2-3 orders of magnitude, for inorganic nitrate associated with aerosol particles in different regions (Ye et al., 2016, 2017a). Aerosol acidity and organic matrix are probably the two major factors in determining the NITs photolysis rate constant. While the authors have attempted to examine the effect of varying NITs photolysis rate constant by using different J(scale) values in the model, the validation through comparison with measurements from only one location at the CVAO is not sufficient. The authors should state explicitly the limitation of this modeling effort and the uncertainty in the global scale extrapolation. To truly understand the global impact of nitrate photolysis in sea-salt aerosol on NOx, OH, and O3 in the MBL, much more field studies in different marine environments are needed.

Response:
Our focus on comparisons with measurements from CVAO is due to the fact that it is a unique dataset due to the availability of concurrent measurements of relevant chemical species ($NO_x$, $HNO_3$, p-$NO_3$, $O_3$, and $HNO_2$) needed to evaluate the model representation of $NO_x$ cycling in the marine boundary layer. Nevertheless, we agree with the reviewer than more field studies in different marine environments are needed to better understand the global impact of nitrate photolysis in sea-salt aerosol. We have noted the limitations of our model study in the second paragraph of Section 4 (page 13, lines 16-20 in the original manuscript). In response to the reviewer's comments, we have modified this paragraph to make this clearer – the modified paragraph now reads:

**We emphasize that our analysis of the impact of NITs photolysis on MBL oxidation chemistry is constrained by measurements only at CVAO. Our focus on CVAO is due to the availability of concurrent measurements of relevant chemical species ($NO_x$, $HNO_3$, p-$NO_3$, and HONO) needed to evaluate the model representation of $NO_x$ cycling in the MBL. Our results therefore point to the need for extending and enhancing the MBL $NO_x$ chemistry measurements at CVAO, and for the need for similar long-term measurements in other marine environments. In particular, accurate measurements of the diurnal variation of HONO at a variety of MBL locations can provide further insight into the impact of NITs photolysis on reactive nitrogen cycling over oceanic regions. From a mechanistic perspective, laboratory and field measurements are needed to better understand the mechanism and rates of ambient p-$NO_3$ photolysis, including the extent to which nitrate photolysis may occur on other aerosol types**

**over both land and ocean regions. Further model developments are also needed to delineate the fraction of nitrate associated with fine-mode sea-salt aerosol and to better couple the calculation of acid uptake on sea-salt aerosols with the MBL halogen chemistry mechanism used in the model. High resolution regional model studies are needed to better understand impacts of nitrate photolysis in continental source regions. Future model studies will be needed to assess the implications of p-NO$_3$ photolysis with regards to interpreting oxygen isotopic composition measurements of nitrate (Alexander et al., 2009; Savarino et al., 2013) and assessing changes in the tropospheric O$_3$ distribution from pre-industrial to present.**

Reviewer 1, comment 2:
Is there a depositional term in the model for HONO, O$_3$ and NO$_2$? The authors did not give any explicit description on deposition parameterization; the answer to the question is probably yes, judging from significant diurnal variations in the modeled HONO concentration. Seawater is a slightly basic solution and is an effective sink for HONO and O$_3$. Since measurements at the Cape Verde Atmospheric Observatory were made only a few meters above the sea level, the depositional loss can significantly affect the observed concentrations near the sea surface, for example, resulting in strong diurnal variations. The depositional loss may explain in part the lower HONO concentrations (Reed et al., 2017; Ye et al., 2017b) at the CVAO than those measured on board the C-130 aircraft (100 – 1000 m above sea level) (Ye et al., 2016). The depositional loss mechanism should be included in the model if it has not been yet.

Response:
Dry deposition terms for O$_3$ and NO$_2$ are included in the current version of the model, but our version of the model does not include a dry deposition term for HONO since this term is poorly constrained. While neglecting this term might have some effect on simulated nighttime HONO concentrations in the MBL, the impact should be negligible in terms of simulated daytime HONO concentrations because the lifetime of HONO during the day is controlled by photolysis and not by dry deposition. Dry deposition velocities for species with Henry's law coefficients similar to HONO are typically 0.5-1 cm s$^{-1}$ over oceans. For a daytime MBL depth of 1 km, this corresponds to a lifetime against dry deposition of about 1-2 days. By contrast, HONO has a lifetime of minutes against photolysis during the day at a location like CVAO (page 9, lines 7-8 in the original manuscript).

We have performed a short 4-month simulation (Sep-Nov 2015) for the $J_{scale}^{100}$ case with added HONO dry deposition, assuming an effective Henry's law coefficient of 10$^5$ M atm$^{-1}$ (Wesely et al., Atmos. Environ., 23, 1293-1304, 1989). The figure below shows a comparison of the diurnal HONO cycle at CVAO during Nov-Dec 2015 from this sensitivity run and from the corresponding simulation with no HONO dry deposition. As expected, adding HONO dry deposition has a minimal effect on simulated daytime HONO surface concentrations at CVAO.

[Figure]

Rather than redo all our simulations, we have added the following text at the end of Section 2.1 to address the reviewer's comments:

**The current version of the model does not include a dry deposition term for HONO. While neglecting this term might have a small effect on simulated night-time HONO concentrations in the MBL, the impact should be negligible in terms of simulated daytime HONO concentrations because the lifetime of HONO during the day is controlled by photolysis and not by dry deposition. Dry deposition velocities for species with Henry's law coefficients similar to HONO are typically 0.5-1 cm s$^{-1}$ over oceans. For a daytime MBL depth of 1 km, this corresponds to a lifetime against dry deposition of about 1-2 days. By contrast, HONO has a lifetime of minutes against photolysis during the day at a location like CVAO.**

**REVIEWER 2**

Reviewer 2, comment 1:
Page 3 Lines 25-30 The authors indicate that sea spray aerosol is generated/transported in two size fractions, and then go on to say that only the nitrate and sulfate associated with 'coarse mode' particles is designated as sea spray SO4S and NITS. Why not also explicitly track this sulfate and nitrate in the smaller particles? Is there some particular reason of is it related the history of the model? What is the consequence of excluding it for this analysis? This comes up again a couple of times and it might be useful to provide a bit more information here.

Response:
This particular aspect of the model is a historic feature arising from the need to consider thermodynamic equilibrium of the total accumulation mode $H_2SO_4$-$HNO_3$-$NH_3$-$H_2O$ system. While nitrate and sulfate uptake by accumulation mode sea-salt is included, the model does not explicitly track these 'sea-salt components' separately from sulfate and nitrate associated with other accumulation mode aerosol types. We note that we already discuss this limitation further on page 4, lines 21-27 in the original manuscript and in the last paragraph of Section 3.1, and we also note the need to explore this issue further in the concluding section (page 13, lines 20-21 in the original manuscript). We have added additional text in response to comment 5 (see below) to discuss the likely impact of adding accumulation-mode SSA nitrate photolysis.

Reviewer 2, comment2:
Page 4 Lines 29-32 What is the rationale for using $J_{HNO3}$ to scale $J_{NIT}$? Given that the rate of photolysis is so different, isn't it also likely that the wavelength dependence is also different? In that case, the scaling factor is likely to change with solar zenith angle. Was this considered?

Response:
Lacking better information, we relied on the empirical finding by Ye et al. (2016) from their analysis of NOMADSS data. That analysis indicated that the missing HONO source correlated well with the product of p-$NO_3$ and the $J_{HNO3}$. To clarify this, we have modified line 29 on page 4 of the original manuscript to read:

**Following the empirical analysis of Ye et al. (2016), the NITs photolysis rate coefficient ($J_{NITs}$) is scaled to the photolysis rate of HNO₃ ($J_{HNO3}$).**

Given the sparsity of information, we did not further explore how the scaling factor itself might depend on variables like the solar zenith angle. To flag this for future study, we have modified lines 18-19 on page 13 of the original manuscript to read:

**From a mechanistic perspective, laboratory and field measurements are needed to better understand the mechanism and rates of ambient p-NO₃ photolysis, including the extent to which nitrate photolysis may occur on other aerosol types over both land and ocean regions.**

Reviewer 2, comment 3:

Page 6, Lines 21-29 Was any particle filter used upstream of the LOPAP? If not, is it possible that some of the signal attributed to gas phase HONO might actually be related to particle phase nitrite associated with SSA? An artefact like this could lead to a strong positive bias in the inferred gas phase HONO measurements, with important implications for the radical budgets. In particular, it could explain why a scaling factor of 25-50 is sufficient to reproduce the NOx measurements at CVAO, but not the 'HONO' measurements.

Response:

We did not use particle filter upstream of the LOPAP, as this would have introduced a potential source of positive artefact - HONO is well known to readily form on surfaces, such as filter media, through dark and photoenhanced processes. To reduce related potential inlet artifacts, we employed as short an inlet as possible by installing the inlet box outside, which is standard practice for LOPAP deployments and mimics the arrangements used in simulation chambers under which the technique has been validated. The dual channel approach of the LOPAP is designed to account for any interferents. Kleffmann et al. (2006) have explicitly shown that particle bound nitrite does not give rise to significant interference in the LOPAP. If such a positive bias were to occur, we would expect this to also be apparent at night – where the observed low HONO concentration (<1 pptv) at night puts a (low) upper limit on any such contribution.

We have added the following text at the end of the first paragraph of Section 2.2.3 to clarify this:

**We note that Kleffmann et al. (2006) have explicitly shown that particle bound nitrite does not give rise to significant interference in the LOPAP. If such a positive bias were to occur, we would expect this to also be apparent at night. The observed low HONO concentration (<1 pptv) at night puts a (low) upper limit on any such contribution.**

Reviewer 2, comment 4:

Section 3.3 and Figure 8 Does the absolute rate of NO$_x$ production from PAN decomposition also change when nitrate photolysis is included? How much of the change in the NO$_x$ production rate is via this indirect impact as opposed to the nitrate photolysis itself? This may be a minor point, but would be interesting to know.

Response:
We assume that the reviewer's question relates to the indirect effect associated with changes in PAN transport and subsequent decomposition to NO$_x$. This is an interesting question. While we archive model-calculated, in situ chemical production rates of NO$_x$ from PAN decomposition and NITs photolysis from each simulation, these archived rates cannot be used to assess the

effect of changes arising from changes in PAN transport alone because of the tight chemical coupling between NO$_x$ and PAN in the MBL (i.e. the diagnosed chemical production from PAN will reflect not only changes in transported PAN, but also changes in in situ production of PAN due to the added NO$_X$ source from NITs photolysis). We can however infer that this indirect effect is small since changes in simulated PAN above the MBL are quite small. We therefore have added the following text in Section 3.3 to address the reviewer's question:

**We further infer that the source of MBL NO$_x$ from long-range transport of PAN does not change significantly when NITs photolysis is included in the model. Above the MBL, maximum increases in PAN concentrations range from 2-5% and 5-10% for the $J_{scale}^{25}$ and $J_{scale}^{100}$ simulations, respectively, relative to the $J_{scale}^{0}$ simulation.**

To more correctly diagnose the relative impact of adding NITs photolysis on NOx chemical production, we modify the middle and bottom panels of Figure 8 to clearly compare NOx source from PAN decomposition in the $J_{scale}^{0}$ run with NO$_x$ production from NITs photolysis in the $J_{scale}^{25}$ and $J_{scale}^{100}$ runs. To reflect this, we have changed lines 15-17 on page 11 of the original manuscript to read:

**This can be understood by comparing the spatial pattern of chemical production rate of NO$_x$ from NITs photolysis in the $J_{scale}^{25}$ and $J_{scale}^{100}$ model runs with the spatial pattern of the chemical production rate of NO$_x$ from PAN decomposition in the $J_{scale}^{0}$ model run, as shown in Fig. 8.**

Reviewer 2, comment 5:
Figure 10 and last paragraph of section 3.3 – would this conclusion of the limited spatial extent of the impacts still be true if the nitrate in the accumulation mode SSA had been considered? It might be worth a sentence or two addressing this here. This is a slightly different question than what is being addressed in Section 3.4.

Response:
We agree with the reviewer that this is worth addressing. We have added the following text at the end of Section 3.3:

**If photolysis of accumulation-mode SSA nitrate had been included, the impacts on MBL NO$_x$ (in terms of relative changes in concentration) would be more wide-spread, especially over remote Southern Hemisphere oceans where a significant fraction of modeled nitrate is in the accumulation-mode (Figure S1). The impact in the vertical will likely still be limited, owing to the relatively short lifetime (~1 day) of accumulation-mode SSA against deposition (Jaeglé et al., 2011).**

Reviewer 2, comment 6:

Section 3.4 The caption for Figure 11 and the associated text are very confusing. What do the asterisks denote – the inclusion of the photolysis scaling factor for all nitrate? That's not clear at all. I can understand why the authors are keen to extend the modelling exercise to probe the impact on nitrate photolysis more generally, but given the strongly non-linear results and the coarse model resolution, to me this section detracts from the rest of the manuscript rather than adding to it.

Response:
We believe that it is important to provide an initial assessment of the potential impact of p-NO$_3$ photolysis on aerosols other than SSA to set the stage for more detailed future model studies. We agree with the reviewer that the coarse model resolution preclude from definitive conclusions in this regard. We have therefore chosen to make it one small part of the paper, and have carefully caveated our conclusions on this aspect of our analysis (page 13, lines 1-5 of the original manuscript).

With regards to Figure 11, the asterisks denote the simulation with added photolysis of accumulation-mode p-NO3. This notation is clearly identified on page 12, lines 11-13 of the original manuscript. To make this clearer, we modify the caption for Figure 11. The new caption for Figure 11 reads:

**Simulated 2014-2015 boundary layer-average (average over bottom 1 km) accumulation-mode nitrate (NIT) and NO$_x$ mixing ratios for the $J_{scale}^{25}$ model (that includes photolysis of only coarse-mode sea-salt nitrate) and the $J_{scale}^{25\,*}$ model (that includes photolysis of both coarse-mode sea-salt and accumulation-mode NIT) runs. NIT and NO$_x$ for the $J_{scale}^{25}$ model are shown in the top left and top right panels, respectively. Middle panels show regions where NIT (NO$_x$) mixing ratios decrease (increase), and bottom panels show regions where NIT (NO$_x$) mixing ratios increase (decrease), due to accumulation-mode NIT photolysis.**